# Numerical assessment of wake-based estimation of instantaneous lift in flapping flight of large birds

**Victor Colognesi** \***, Renaud Ronsse, Philippe Chatelain**

Institute of Mechanics, Materials and Civil engineering, UCLouvain, Louvain-la-Neuve, Belgium

\* victor.colognesi@uclouvain.be

## Abstract

Experimental characterization of bird flight without instrumenting the animal requires measuring the flow behind the bird in a wind tunnel. Models are used to link the measured velocities to the corresponding aerodynamic forces. Widely-used models can, however, prove inconsistent when evaluating the instantaneous lift. Yet, accurately estimating variations of lift is critical in order to reverse-engineer flapping flight. In this work, we revisit mathematical models of lift based on the conservation of momentum in a control volume around a bird. Using a numerical framework to represent a flapping bird wing and compute the flow around it, we mimic the conditions of a wind tunnel and produce realistic wakes, which we compare to experimental data. Providing ground truth measurements of the flow everywhere around the simulated bird, we assess the validity of several lift estimation techniques. We observe that the circulation-based component of the instantaneous lift can be retrieved from measurements of velocity in a single plane behind a bird, with a latency that is found to depend directly on the free-stream velocity. We further show that the lift contribution of the added-mass effect cannot be retrieved from such measurements and quantify the level of approximation due to ignoring this contribution in instantaneous lift estimation.

## 1 Introduction

Bird flight has been a source of inspiration for ages. While modern aerodynamics leads to better understanding of the mechanisms behind flapping flight, it is still not perfectly understood. The impressive performance of the bar-tailed godwit—which has been found to migrate for 11000 km without stopping or feeding [1, 2]—is but an example, among others, of the efficiency of bird flight. It is thus of great interest for both biologists and engineers to observe and study birds to further understand the complex mechanisms governing their flight.

The study of bird flight, however, involves many challenges. Collecting experimental data on the aerodynamic forces produced by a bird in flapping flight without hindering it with sensing instruments, often relies on measurements in the wake of the animal. PIV (Particle Image Velocimetry) allows such measurements by measuring the velocity of the fluid on a surface in

**Data Availability Statement:** The 3D velocity and vorticity fields collected in the wake of the bird for experiment 1 are available at https://doi.org/10.14428/DVN/KQQAQJ. 2D velocity and vorticity fields collected at three downstream distances

during experiment 2 are available at https://doi.org/10.14428/DVN/7ATEGZ.

**Funding:** Victor Colognesi is supported by a FRIA grant (Grant number FC 21291) from the Fonds de la Recherche Scientifique de Belgique (F.R.S.-FNRS, https://www.frs-fnrs.be/en/). This work is part of a project supported by the ARC program (grant number 17/22-080, RevealFlight) of the Federation Wallonie-Bruxelles (https://www.federation-wallonie-bruxelles.be). Computational resources have been provided by the Consortium des Equipements de Calcul Intensif (CECI), funded by the Fonds de la Recherche Scientifique de Belgique (F.R.S.-FNRS) under Grant No. 2.5020.11 and by the Walloon Region. The present research also benefited from computational resources made available on the Tier-1 supercomputer of the Federation Wallonie-Bruxelles, infrastructure funded by the Walloon Region under the grant agreement 1117545. The funders had no role in study design, data collection and analysis, decision to publish, or preparation of the manuscript.

**Competing interests:** The authors have declared that no competing interests exist.

the wake, like in [3]. Models are then required to compute the aerodynamic forces from these velocity fields in the wake.

These models can be based on various assumptions. In the works of Spedding et al. [4] and Henningsson et al. [5], for example, the spanwise vorticity is evaluated (using PIV in a transversal plane). More precisely, [4] relies on simplified models to link the vorticity to lift values, where the wake is considered to be composed of distinct vortices forming either two deforming tip vortices of constant circulation, or vortex rings. In [5], only the variations of lift through a flapping cycle are considered, since the spanwise vorticity is linked to the temporal variations of the wing circulation. Stalnov *et al.* [6] and Nafi *et al.* [7] use a similar approach to estimate the lift of a bird, but limited to a single plane of measurement in its wake, which cannot capture the spanwise variations of the wing circulation.

Other works consider the streamwise vorticity, that can be measured in cross-flow planes (also called Trefftz planes). This component of the vorticity is shed from the wing trailing edge as its circulation varies along the span. Since the circulation reaches zero at the wingtips, any lift-producing wing will thus leave a wake with a streamwise component of vorticity. Measuring this component enables other methods for the estimation of the lift. In the works of Henningsson et al. [8] and Muijres et al. [9], lift variations are estimated by identifying distinct tip, root, and tail vortices. Other works such as the ones of KleinHeerenbrink et al. [10], Johansson et al. [11] or Hedh *et al.* [12] use the complete vorticity field to compute average lift in gliding [10] or flapping [11, 12] flight.

While multiple methods exist to compute the lift of a bird from wake measurements, it has been shown by Gutierrez et al. [13] that commonly used methods do not necessarily agree with each other and that their results can be dependent on the streamwise distance between the bird and the measurement plane. While most experimental measurements of bird wakes are achieved in wind tunnels where the distance between the bird and the measurement plane is constant, in [13], the bird flies through a fixed measurement plane in still air. This experiment highlights the deformations occurring in the wake and shows that they can affect the result of commonly used lift estimation methods. It also means that the wake of a bird is not necessarily frozen, although thus is often assumed.

Results of [13] could be caused by vortex breakdown, as they hypothesize, or by non-zero streamwise velocities, as suggested in [11]. In either case, 3D data is necessary to further understand the evolution of the wake and how to use it in order to estimate the lift of the bird producing it. Numerical frameworks are useful tools to generate 3D velocity fields in the wake of a bird, which would be difficult to measure experimentally. In [14], Wang et al. simulated the flow around and behind flapping wings to evaluate the validity of a lift estimation method based on the application of the Kutta-Joukowski theorem, using the streamwise component of the vorticity field in cross-flow planes. They propose a correction of the method that enables an accurate estimation of the average lift, but their proposed estimation fails to recover the unsteady lift of a flapping wing.

In the present paper, we use an in-house model of a flapping bird [15] based on a vortex particle-mesh method [16], which can faithfully simulate the unsteady wake of a bird over large distances. We use simulation results to analyze the unsteady wake following control-volume and outflow-plane approaches, as illustrated in Fig 1. The former considers integrals of the velocity and vorticity fields in a closed surface around the bird to compute the aerodynamic forces it produces. The latter aims at estimating the aerodynamic forces from measurements limited to a single cross-flow plane in the bird wake.

We apply this framework to the flapping flight of a large migratory bird. This case is characterized by a higher Reynolds number and a lower reduced frequency than in [14]. As both of these parameters have a significant impact on the wake dynamics and the wing aerodynamics,

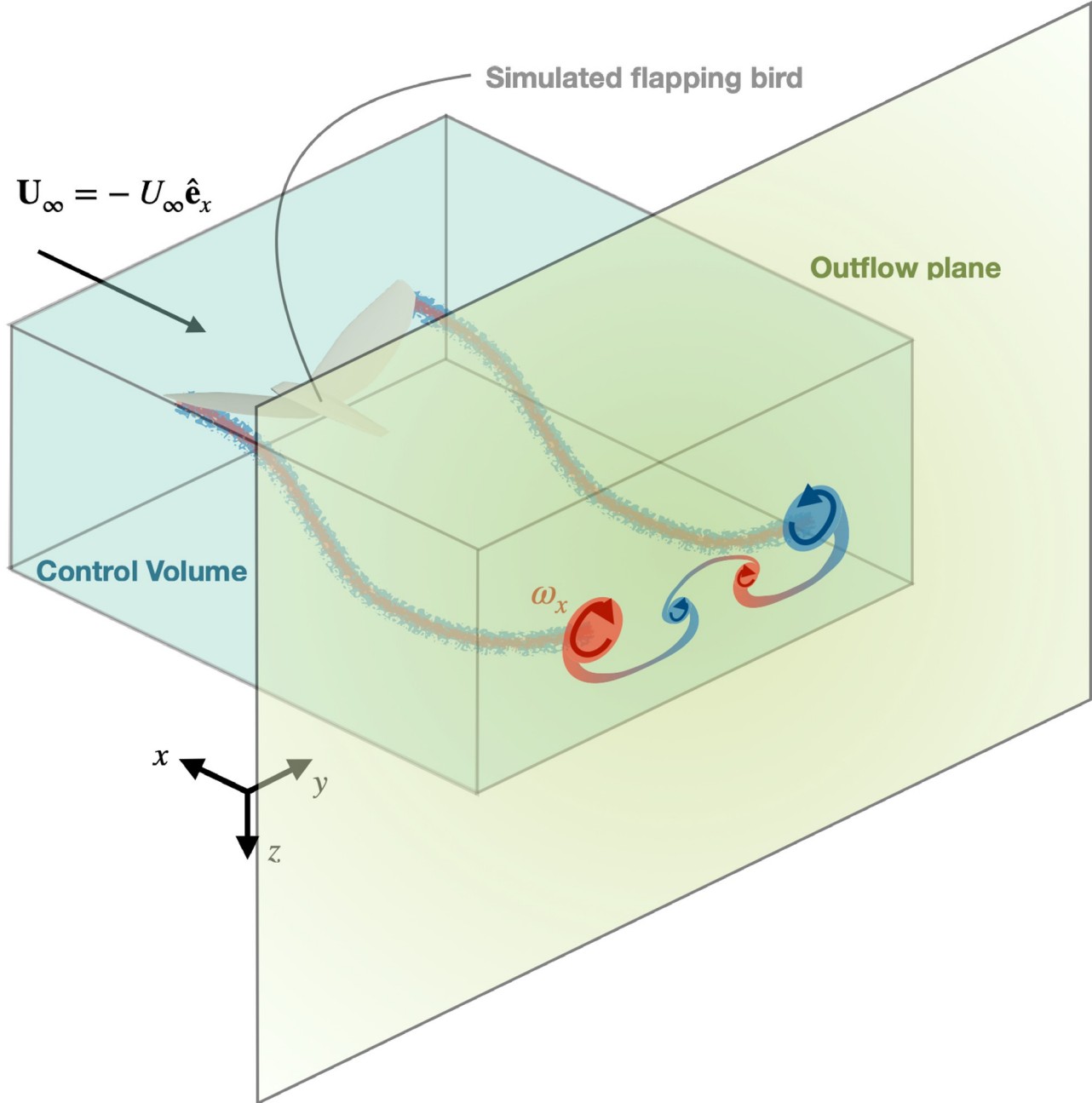

**Fig 1. Control volume around a bird and outflow plane in its wake.** The tubes behind the bird give a simplified illustration of the tip vortices of the bird, and the red and blue shapes in the outflow plane illustrate the streamwise component of the vorticity field, with the color corresponding to its sign. The axes of the reference frame used throughout the paper are also represented.

we expect our results to differ from the ones of [14]. In particular, the Reynolds number is linked to the importance of viscous effects in the flow: at higher Reynolds number, viscous effects are less important than inertial ones, and thus, wake dissipation is slower. The reduced frequency is directly related to the relative importance of the added mass contribution to the lift compared to the one of the circulation (see S3 Appendix). For lower values of this parameter, we expect the aerodynamic forces to be dominated by circulatory lift, which leaves a trace

in the wake behind the bird [4, 9]. It is worth testing whether, under these conditions, the unsteady lift can be accurately estimated from wake measurements.

In sum, we propose a derivation of an expression allowing us to estimate the instantaneous lift of a bird in a flapping regime from velocities measured in the wake. At every step of the development, we highlight the assumptions that lead to a formula commonly used to compute the lift from such data. We then assess the validity of the derived formulation in numerically produced wakes using our in-house flapping bird model. Although our results come from simulations, that rely on many assumptions and do not represent exactly the reality of a bird wake, the main goal is to establish the link between the aerodynamic forces and the resulting wake velocities. Despite the differences with actual bird wakes, the validity of the formulation can be assessed as long as the wake dynamics are accurately captured and the simulated wake presents the same features as a real one. To verify this last point, we compare our results to experimental measurements of the wake of a swift—i.e., a bird having also a high aspect ratio—found in [5, 8].

## 2 Materials and methods

### 2.1 *In silico* wind tunnel

The simulation framework used to produce the results is based on the model presented thoroughly in [15]. It consists of a bio-mechanical model of a bird skeleton coupled to a vortex particle-mesh-based fluid solver. The motion of the bird and its wings are computed using the bio-mechanical model, and the fluid solver computes the resulting aerodynamic forces applied to the feathers of the bird.

Although the model presented in [15] is applicable to many flapping birds of high aspect ratio, the bird simulated here is a Northern Bald Ibis (*Geronticus Eremita*) with a mass of 1.2 kg and flying at a velocity of $U = 15\text{ms}^{-1}$. Its flapping frequency $f$ is 4 Hz and the wavelength of the wake $\lambda = U/f$ is thus equal to 3.75 m. To mimic the conditions of a wind tunnel, the simulation domain is subjected to a constant inflow at a velocity of $U_\infty = 15\text{ms}^{-1}$, ensuring that the bird remains at a fixed position in the simulation frame. With a mean aerodynamic chord of the wing $\bar{c} = 0.18$ m, we define the Reynolds number $Re = U_\infty \bar{c}/\nu = 1.8 \ 10^5$ and reduced frequency $k = \omega \bar{c}/2U_\infty = 0.15$, where $\nu$ is the kinematic viscosity of the fluid ($\nu = 1.48 \ 10^{-5}\text{m}^2\text{s}^{-1}$ for air) and $\omega = 2 \ \pi f$.

Fig 2 pictures the various models used to represent the bird in the simulation, and a photograph of the Ibis to provide a point of comparison. Panels of this figure will be further used to support the description of these models throughout this section.

**Bio-mechanical model.**   The bio-mechanical model of the bird wing is extensively described in [15]. It is composed of three rigid segments corresponding to the main bones of the wing, as represented in panels (B) and (C) of Fig 2. These segments are articulated together and to a body-tail ensemble, which does not deform.

The model is implemented using ROBOTRAN [17], a software handling the dynamics of multi-body systems. In ROBOTRAN, a separate frame is defined after each joint rotation. The rotation angles are thus defined in the local frame. In our model, the shoulder joint has three rotations, the elbow is rotating about the $y$ axis only, and the wrist rotates about three successive orthogonal axes.

The plumage of the bird is represented using a reduced set of control-feathers. These are attached to the bones directly, through spring-like joints. Additionally, the feathers are coupled to each other with spring-like components, ensuring a smooth spreading of the plumage when the skeleton moves. Although they are rigid, the feathers move when the skeleton is actuated or when they are subjected to aerodynamic forces.

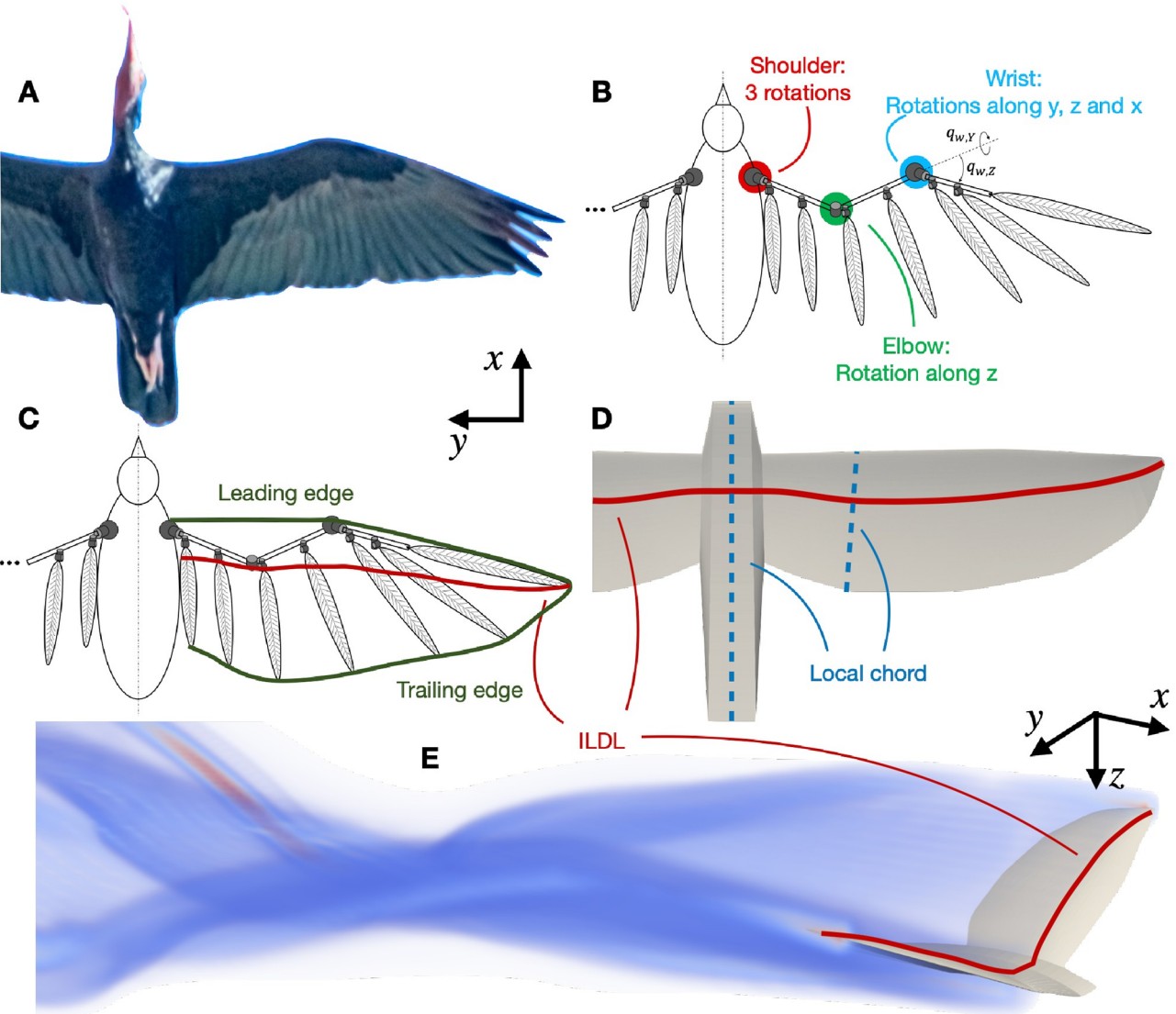

**Fig 2. Various models are used to capture the bird in the simulation framework.** (A): Picture of a real bird while gliding (Original photograph from Peter W. Hills, reproduced with permission). (B) bio-mechanical model of the skeleton and feathers, with the actuated joints being highlighted and two rotations of the wrist being represented as examples. (C) Immersed Lifting and Dragging Line (ILDL) obtained in a particular configuration. (D) final planform represented in the fluid solver. (E) simulated bird and its near wake.

**Bird aerodynamics.** The bird aerodynamics are computed through an Immersed Lifting and Dragging Line method (ILDL) [18]. The position of this line is obtained from the current position of the bones and feathers embodied into the bio-mechanical model. As shown in panel (C) of Fig 2, the leading edge of the wing is defined as the line starting from the shoulder, going straight to the wrist joint (thus following the propatagial tendon), and then following the hand segment and the last primary feather. The trailing edge links the tip of the feathers, from the last secondary (the closest to the body), to the last primary (at the wingtip). The ILDL is then defined as the line passing through the quarter of the local chord (i.e. the line connecting the leading and trailing edges) at each point along the wing, and normal to the chord.

At each section of the wings, the lift and drag are computed based on the local instantaneous relative velocity, using polar data, i.e., the values of the lift and drag coefficients of a 2D

profile as a function of both the local angle of attack and Reynolds number. The lift and drag coefficients at each point of the ILDL are those of an AS6092 airfoil [19]. It is a bird-like airfoil, and its polar is available in [19] and represented in S1 Appendix. This particular airfoil has been chosen among the proposed airfoils of [19] for its limited camber and intermediate thickness. The polar of the airfoil is computed in [19] using XFOIL [20]. The aerodynamic forces computed along the ILDL take in consideration the effects of local chord, angle of attack, thickness and camber through the polar.

The use of a single airfoil for the whole bird wing is a major assumption, since camber and relative thickness realistically vary along the span of the bird. However, we believe that the aerodynamic forces computed with this assumption remain realistic—although they do not correspond exactly to the ones that the bird would produce. As our thesis rather lies in proving the recovery of aerodynamic forces produced by the bird from wake measurements, a slight lack of fidelity in the generation of these forces does not undermine that core message, as long as they resemble the ones produced by an actual bird.

The body-tail ensemble is included in the ILDL. Between the wings, the chord length is set to the length of the bird body. In this region, the aerodynamic performance of the airfoil is degraded in regarding the lift-to-drag ratio. To account for the larger contribution of the body to the drag of the bird than to its total lift, we use a degraded lift coefficient compared to the rest of the ILDL. When computing the lift coefficient of the body, we divide the value obtained from the polar by 3 in the simulations, while keeping the same drag coefficient. This value roughly corresponds to the increase of chord length between the wing root and the body, and thus reduces the spanwise variation of the circulation at that location. This is an arbitrary value, but the main source of aerodynamic forces being the wings of the bird, we assume that this should have a minor effect on the results.

The lift contribution of the body-tail ensemble in real birds is still a matter of open discussion. In [21], it is shown that the body-tail of gliding raptors have a higher circulation than the wing at their root, and thus produces more lift. However, in other experimental results comparing the wakes of flapping birds with and without a tail [22], no evidence was found for an aerodynamic effect of the tail. Finally, in [8], the effect of the tail was found to significantly vary with the flight velocity, with higher (resp. lower) lift produced at lower (resp. higher) velocities, which is consistent with the observations of [23], who observed that birds tend to furl their tails in fast flight. Finally, it is shown in [24] that the cost of transport of large flapping birds increases with the opening angle of their tails, and that reducing tail lift thus improves flight efficiency. Since our scenario concerns steady flapping flight at a high forward velocity, for a large bird with a high aspect ratio, we consider that the assumption of low tail lift is reasonable.

The simulation framework does not currently compute the added mass effect on the wing and the related aerodynamic forces. This phenomenon being due to potential effects, it mostly affects the flow directly around the wing, and it does not influence the flow further behind. However, it can influence the evolution of the vortical structures in the wake shortly after their shedding. The added mass contribution to the aerodynamic force can be computed *a posteriori* from the wing kinematics. To evaluate this force, we consider each wing profile to behave like an ellipse with a length equal to the chord $c$ and a width equal to the thickness $t$ of the profile. The acceleration of the profile is obtained from the simulation, and the local force is computed as if it was a 2D flow. For the three degrees of freedom of the 2D profile (i.e., the vertical and horizontal translations and the pitching rotation), the added mass forces are then expressed as

$$\mathbf{F}_{am} = -M_a \mathbf{a} \ , \tag{1}$$

where $\mathbf{F}_{am} = [F_z, F_x, M_y]^T$ is the added mass force vector expressed in the local 2D frame, $\mathbf{a} = [\ddot{z}, \ddot{x}, \ddot{\theta}]^T$ is the 2D acceleration vector and $\mathbf{M}_a$ is the added mass matrix

$$\mathbf{M}_a = \rho \begin{pmatrix} \dfrac{\pi c^2}{4} & 0 & -\dfrac{\pi c^3}{16} \\[3mm] 0 & \dfrac{\pi t^2}{4} & 0 \\[3mm] -\dfrac{\pi c^3}{16} & 0 & \left(\dfrac{c^2}{4} - \dfrac{t^2}{4}\right)^2 \end{pmatrix}. \tag{2}$$

The matrix $\mathbf{M}_a$ of Eq (2) was obtained using the coefficients provided for an ellipse in Table.4.3 of [25] (using $a = c/2$ and $b = t/2$), and modifying them so that the moment is computed about the quarter of the chord.

**Solving the unsteady wake flow.** The flow in the bird wake is computed by solving the incompressible Navier-Stokes equations in their vorticity-velocity ($\omega$–$\mathbf{u}$) formulation. This is done through an hybrid vortex particle-mesh method, where the vorticity field is discretized into a set of particles, that are advected using the velocity field. The differential operations in the equations are solved through fourth order finite differences on a cartesian grid. High order interpolation schemes are used to pass the information back and forth between the particles and the grid at each time step. This implementation is done using the PPM library described in [26].

The mesh is also used to *reset* the particles on grid points every 5 timesteps to prevent clustering or depletion of particles in some areas. Finally, particle discretization does not ensure that the vorticity field remains divergence-free. To prevent errors related to this, a reprojection operation is performed on the vorticity field every 20 timesteps. Both of these operations are described in details in [16].

The ILDL representing the bird acts as a source of vorticity in the fluid solver and vortical structures related to the production of lift and drag are shed from it. Vortex particles are emitted to represent the shedding due to both the variations of circulation along the line and through time. In addition, vortex dipoles are shed to represent the signature of drag in the wake. More detail on these methodological aspects can be found in [18]. Panel (E) of Fig 2 shows the intensity of the vorticity field behind the ILDL of a representative simulation scenario.

The momentum in the wake has been confirmed to correspond to the actual aerodynamic forces produced by a simulated bird in [15], and for fixed wings in [18]. This correspondence is key in our goal of deriving expressions for the estimation of lift from wake measurements. The model used to simulate the wing aerodynamics, the shedding of vorticity and the wake evolution has been compared to experimental data in the challenging case of rotorcraft in [27], where the aerodynamic forces and their variations with the control parameters of the rotor were accurately modeled. The coupled bio-mechanical and aerodynamic model used to simulate bird flight have been compared to reference results regarding flight power in [15].

**Actuation of the model and flight controller.** The wing kinematics used in the simulations result from seven joint rotations in the wing skeleton, represented in panel B of Fig 2. Each joint angle $q_i$ follows a pure sinusoidal trajectory, governed by a unique frequency $f = 4$ Hz, and joint dependent mean value $q_{0,i}$, amplitude $A_i$ and phase $\varphi_i$. Each joint $i$ thus follows a trajectory defined as $q_i(t) = q_{0,i} + A_i \sin(2\pi ft + \varphi_i)$. The parameters $q_{0,i}$, $A_i$ and $\varphi_i$ are adjusted in order to resemble experimentally observed bird wing kinematics. The derivation of the

**Table 1. Kinematic parameters used in the simulations for the right wing (the left wing is symmetric).**

|  | Shoulder $Y$ | Shoulder $X$ | Shoulder $Z$ |
|---|---|---|---|
| $q_{0,i}$ [deg] | 0.63 | 0 | 9.59 |
| $A_i$ [deg] | 2.14 | 22.22 | 11.11 |
| $\varphi_{0,i}$ [rad] | $-\pi/2$ | $\pi$ | $-\pi/2$ |
|  | Elbow $Z$ | Wrist $Y$ | Wrist $Z$ | Wrist $X$ |
| $q_{0,i}$ [deg] | $-16.67$ | 0 | 16.67 | 0 |
| $A_i$ [deg] | 16.67 | 1.85 | 16.67 | 0 |
| $\varphi_{0,i}$ [rad] | $\pi/2$ | $-\pi/2$ | $-\pi/2$ | N.A. |

parameters is reported in [15] for the same species, using experimental data from [28]. The resulting values of the kinematic parameters are provided in Table 1.

Closed loop controllers are used to stabilize and trim the flight. That is, the bird adapts the trajectories of certain joints to ensure that it stays at equilibrium and remains on average at the same velocity and altitude. Four parameters are selected for this purpose: the mean values of the pitching and sweeping angles of the shoulder joint ($q_{0,s,\ Y}$ and $q_{0,s,\ Z}$) and the amplitude of the pitching and flapping motion of the shoulder joint ($A_{s,Y}$ and $A_{s,X}$). The values of these parameters are determined through the use of PID controllers taking as input the period-averaged values of the altitude, forward flight velocity and pitch angle and their reference values. The wing kinematics and the stabilization of the *in silico* bird are presented in more detail in [15].

## 2.2 Derivation of a lift estimation model

**General expression of the aerodynamic force acting on a flying body.** The aerodynamic force acting on a flying object—a bird in this case—results from an exchange of momentum between it and the fluid. The conservation of momentum in a control volume around the bird allows to express the aerodynamic force from the velocity and pressure of the fluid in and around this volume. Indeed, the aerodynamic force $\mathbf{F} = [F_x, F_y, F_z]^{\mathrm{T}}$ acting on an impermeable body in a static volume $V$ bounded by a surface $S$ can be expressed as

$$\frac{\mathbf{F}}{\rho} = -\frac{\mathrm{d}}{\mathrm{d}t}\int_V \mathbf{u}\,\mathrm{d}V + \oint_S \hat{\mathbf{n}} \cdot \left[ -\frac{p}{\rho}\mathbf{I} - \mathbf{u}\mathbf{u} + \mathbf{T} \right]\mathrm{d}S \ , \tag{3}$$

where $\mathbf{u} = [u_x, u_y, u_z]$ is the local velocity vector at every point inside of the volume, $\hat{\mathbf{n}}$ the normal to the surface, $p$ the pressure at every point on the surface $S$, $\rho$ the fluid density, $\mathbf{I}$ the unit matrix and $\mathbf{T} = \mu(\nabla\mathbf{u} + \nabla\mathbf{u}^{\mathrm{T}})$ is the viscous stress tensor, with $\mu$ being the dynamic viscosity of the fluid.

The interpretation of Eq (3) is straightforward. The force resulting from exchanges of momentum with the fluid corresponds to the increase of momentum in such fluid, which is the first integral of Eq (3). If the volume is finite, however, the momentum of the fluid inside it is affected by fluxes of momentum at its boundaries and by external forces acting on the same boundaries. This has to be corrected when evaluating the aerodynamic force and corresponds to the second integral of Eq (3).

Although Eq (3) provides a simple and exact expression of the aerodynamic force, it requires access to the velocity field in an entire volume and to the pressure on a closed surface, both of which can prove difficult in real-life experiments. Instead, it is preferable to rely on integrals of the velocity and its derivatives on surfaces around the bird, and ideally on a single

surface behind it called the *outflow* plane. Example of such a control volume and a candidate outflow plane are shown in Fig 1.

Taking Eq (3), Noca et al. [29] used vector calculus and the Navier-Stokes equations to obtain expressions of the instantaneous aerodynamic force a flier is experiencing using only integrals of the velocity and its derivatives on surfaces around it, thus getting rid of the pressure in the expression of the force. This resulting "flux equation" of [29] can be re-written as

$$
\begin{aligned}
\frac{\mathbf{F}}{\rho} = & \oint_S \hat{\mathbf{n}} \cdot \left( \frac{1}{2} u^2 \boldsymbol{I} - \mathbf{uu} \right) \mathrm{d}S + \frac{1}{2} \oint_S \mathbf{x} \times (\hat{\mathbf{n}} \times (\mathbf{u} \times \boldsymbol{\omega})) \, \mathrm{d}S \\
& - \frac{1}{2} \frac{\mathrm{d}}{\mathrm{d}t} \oint_S \hat{\mathbf{n}} \cdot [(\mathbf{x} \cdot \mathbf{u})\boldsymbol{I} - \mathbf{xu} + 2\mathbf{ux}] \, \mathrm{d}S \ ,
\end{aligned}
\tag{4}
$$

where $\boldsymbol{\omega} = \nabla \times \mathbf{u}$ is the vorticity vector. To derive Eq (4) from the "flux equation" of [29], it is assumed that the flow is incompressible and that the contribution of the viscous terms to the estimation of aerodynamic forces is negligible. Since the case that we study is set at a high Reynolds number ($Re = 10^5$) and at a low Mach number, these assumptions are reasonable. The time-dependency of the variables of Eq (4) is omitted for the sake of clarity. Unless stated otherwise, any field or variable will be considered dependent on time in the rest of this paper.

Eq (4) provides an expression of the aerodynamic force that only depends on the velocities on a closed surface around a flying object. While this is more achievable than volume integrals, it is still challenging to access the velocity all around a flying bird. In this work, we focus only on the vertical component of the force, which, on average, compensates for the weight and corresponds to the lift force in level flight. To further simplify the equations and develop an expression of the lift that can be evaluated using experimental measurements, more hypotheses are required. In the following sections, we thus make further assumptions on the velocity field in the bird wake and how they can be used to simplify Eqs (3) and (4) and apply them to the lift estimation.

**Assumptions on the velocity field in a bird wake.** The velocity field resulting from the lift forces acting on the bird can be predicted in a simplified way, especially concerning the scaling of the velocity field when moving away from the bird and its wake. The measured velocity is induced by two lift contributions: the wing circulation and the added-mass effect.

The effect of the wing circulation in the wake can be roughly represented by a horseshoe vortex. Reality is more complex, and even if the vorticity takes the shape of a horseshoe, its intensity and width will vary with the position. But this simple representation allows us to build intuition about the trends of the velocity field. This vortex structure induces a flow that is mostly two-dimensional in the wake and with a velocity magnitude decaying with $1/r^2$ if $r$ is the distance in the $y - z$ plane to the center of the wake.

Any integral of the velocity on a closed surface surrounding the bird is divided into three parts: an integral on a plane normal to the free stream velocity in the wake (also called outflow plane), an integral on the surfaces surrounding the wake, parallel to the free stream velocity, and an integral on a plane normal to the free stream velocity upstream from the bird (called inflow plane). The integral on the outflow plane is necessarily non-zero because the vortex pair crosses the surface, inducing non-zero velocities in it. An integral of the velocity on the lateral planes tends to zero as the volume extends in the lateral directions because of the quadratic decay of the velocity. However, an integral of the velocity multiplied by a distance to the vortex wake ($y$ or $z$) can converge to a non-zero value because the size of the integration surface also increases with $r$. An integral of the velocity on the inflow plane also tends to zero as the distance between the plane and the bird increases because of the absence of vorticity upstream from the bird.

The velocity field related to the added mass effect can be modeled by the flow induced by a set of source-sink dipoles situated on the wing. The intensity of these dipoles is related to the time-varying velocity of the wing. The velocity induced by a single source or sink decays with $1/r^2$, but this is changed to $1/r^3$ in the case of a dipole (with $r$ being the distance to the center of the dipole in the 3D space). The same applies to the finite set of dipoles when evaluating the velocity at a large distance much larger than the span of the wing.

When integrating the velocity induced by the added mass, there is no reason to separate the outflow plane from the lateral ones. The integral over a closed surface around the bird thus tends to zero as the volume expands unless the velocity is multiplied by the distance to the bird ($x$, $y$ or $z$), in which case the integral can converge to a non-zero value.

**Lift estimation from an outflow plane–theory.** From the observations made in the previous section, we can see that the first term of Eq (4) is only non-zero on the outflow plane, provided that the control volume extends far enough in the lateral directions. These integrals only show a contribution from the added mass if the plane is very close to the bird. Since the vorticity field is limited to the vortical structures in the wake, the second integral of Eq (4) can only be non-zero on the outflow plane.

The third integral of Eq (4) contains contributions from both sources of lift (circulation and added mass). The contribution of the lateral planes to this integral cannot be neglected because of the multiplication of the velocity **u** with the position **x**.

Comparing Eqs (3) and (4), we retrieve the momentum flux in both equations. The time-derivative integral of Eq (3) is equivalent to the last term in the last integral of Eq (4). This means that all the remaining terms in Eq (4) correspond to the effect of pressure on the boundaries of the control volume.

An expression of the lift is obtained through the scalar product of the force vector **F** by a unit vector in the vertical direction $-\hat{\mathbf{e}}_z$. Since the pressure only contributes to the lift on the horizontal planes of the control volume, it is possible to build a control volume that extends far enough in the vertical direction so that this term tends to zero. In this case, the only remaining terms correspond to the variation of momentum in the volume and the momentum flux at its boundaries.

Since the volume integral of Eq (3) cannot be computed in experimental setups, the only way to measure its contribution is through its expression in Eq (4). However, it may also prove difficult to compute this integral on planes all around a bird. This is one of the reasons why most experimental works are based on the momentum flux measured in a single outflow plane located behind the animal. As stated in the previous section, the integral for the momentum flux tends to zero for all the other planes, so integrating on a large outflow plane should accurately represent the actual value.

Estimating the lift based on the momentum flux in the wake implies the assumption that the momentum travels in the wake at a steady speed. A measurement at a time $t$ at a distance $x$ behind a bird flying at a velocity $U_\infty$ then corresponds to the lift of the bird at a time $t - x/U_\infty$. However, for this to be true, all the momentum injected into the fluid at that precise time has to exit through the plane at the same moment. This condition corresponds to the frozen-wake hypothesis. This will not be the case if either the distance over which the momentum has to be transported varies (if the wing has a significant amount of sweep) or if the speed at which the momentum is transported varies in the wake.

After developments reported in S2 Appendix, the momentum flux term can be expressed as

$$\frac{L}{\rho} = \frac{\mathbf{F} \cdot (-\hat{\mathbf{e}}_z)}{\rho} = -U_\infty \int_{S_{out}} y\,\omega_x\,\mathrm{d}S - \int_{S_{out}} (u_x + U_\infty)u_z\,\mathrm{d}S \ , \tag{5}$$

where $U_\infty$ is the magnitude of the inflow velocity vector (aligned with $-\hat{\mathbf{e}}_x$). Contrarily to what the signs might indicate, the expression $u_x + U_\infty$ represents departures from the free stream velocity.

The first term of Eq (5) corresponds to a classical fixed-wing formula that can also be derived from lifting line theory and is usually associated with the Kutta-Joukowsky theorem. It will be called the *KJ* term. The second term results from the deficit or surplus of the streamwise component of the velocity and its effect on the transport of momentum, involving the difference $u_x + U_\infty$. It will be called the *velocity departure* or *VD* term.

The developments leading to Eq (5) require the outflow plane to extend to infinity. This means that the lateral and horizontal planes of the control volume are positioned infinitely far from the bird. The contribution of these planes to the integral of the pressure and to the momentum flux, therefore, becomes negligible. As seen before, though, the integral corresponding to the time derivative does not tend to zero as these planes go to infinity.

While an infinite outflow plane is required to evaluate all the velocity in it, it is not the case for the vorticity, which forms a compact field. Since vorticity is contained in a region that does not extend far in the $y$ and $z$ directions, any plane encompassing this region will correctly evaluate the *KJ* term of Eq (5). The *VD* term consists in the product of two terms tending to zero quadratically as $y$ and $z$ increase. This integrand thus quickly converges to zero when evaluated out of the vortical structures of the wake but cannot be considered compact.

In the case of periodic wing kinematics (and thus forces), averaging Eqs (3) or (4) over a flapping period cancels the time-derivatives terms. Therefore, the average of the momentum flux results in the average force if the effect of the pressure on the boundaries of the control volume is neglected, which is an acceptable assumption for a large control volume.

The expression given in Eq (5) can be evaluated using measurements of the velocity in a single, sufficiently large outflow plane behind the bird. From the developments of this section, we see that it enables the estimation of both the average and the unsteady circulatory lift of a bird. The next section describes two numerical experiments designed to verify the validity of this final model provided by Eq (5).

### 2.3 *In silico* scenarios testing the lift estimation models

Two scenarios are reported in this paper, namely scenario 1 and scenario 2. In both, the ground-truth velocity and vorticity fields are available over the whole simulation domain.

In scenario 1, the bird displays perfectly periodic wing kinematics guaranteeing stationary flight. That is, the total forces and moments acting on the bird average to zero over the flapping period. This *in silico* experiment is used to evaluate how accurately the expression of the lift of Eq (5) approaches the actual lift of the bird. Eq (5) is evaluated for outflow planes at various distances downstream from the bird.

We use the data collected in scenario 1 to evaluate the average of the lift estimation as a function of the position of the outflow plane. It is computed through a time integration with the trapeze method of the signal obtained with Eq (5). This allows evaluating the contribution of each term of Eq (5) to the average vertical force and their relevance when one is interested in the average weigh support.

As stated in section 2.2, the estimation provided by Eq (5) is delayed with respect to the instantaneous lift. The time delay between the output of Eq (5) is evaluated for this periodic flight. At each downstream position, it is computed by shifting the estimation provided by Eq (5) through time with a delay $\Delta t$ comprised between 0 and the flapping period. For each value of $\Delta t$, the shifted result is correlated to the actual instantaneous lift function, and the value of $\Delta t$ maximizing this correlation is selected. To the ensemble of the best values of $\Delta t$ at each

downstream distance $x_{out}$, we finally apply a linear regression in order to get a coefficient $a$ capturing this relationship as

$$\Delta t = a x_{out} \ . \tag{6}$$

The computational domain used in scenario 1 extends from -3 m to 3 m in the spanwise and vertical directions (which corresponds to 2.2 spans). However, unbounded boundary conditions are applied, so that the flow is not affected by the boundaries of the domain. In the streamwise direction, the inflow is at a distance of 1 m of the bird (0.74 spans), and the outflow is at a distance of 15 m (11.1 spans, or 4 wavelengths). The resolution is of 64 particles per meter, or 86 per span.

In scenario 2, the bird uses kinematics that lead to non-periodic lift. We use the data collected in this experiment to test the validity of the lift estimation when the period-averaged lift varies.

To obtain the kinematics, we use a closed-loop controller stabilizing the bird. Details about this controller are reported in [15]. After 5 seconds of level flight, the bird is tasked to change its lift so that it follows the following equation

$$L_{\text{targ}} = L_0 \sin(2\pi(t-5)/10) \ , \tag{7}$$

where $L_0 = 5$ N is an arbitrary force. The lift thus varies for 10 seconds. After those 10 seconds, the controller determines the target lift in order to stabilize the bird at a constant altitude. During the manoeuver, the controller also ensures that the bird remains at a constant streamwise velocity.

This results in variations of the lift, which is no longer periodic. The estimations are obtained in this case with the KJ term only, and shifted through time using $\Delta t = (x_{out} - x_{bird})/U_\infty$, where $x_{out}$ and $x_{bird}$ are the respective positions of the outflow plane and the bird. While the outflow plane is fixed, the bird can move through the domain. Its position should be evaluated at a time $t - \Delta t$ to ensure that the delay is correctly evaluated. However, it constitutes an implicit equation. In this scenario, we compute $\Delta t$ with a position evaluated at a time $t - \Delta t_0$, where $\Delta t_0$ is obtained using the initial position of the bird.

Scenario 2 uses a domain with slightly different dimensions as compared to scenario 1. It has the same size in the spanwise direction, extends from -5 to 3 m (-3.7 to 2.2 spans) in the vertical direction, and -2 to 10 m (1.5 to 7.4 spans) in the streamwise direction. Is has a lower resolution than scenario 1, with 27 particles per span.

## 3 Results

### 3.1 Aerodynamics and simulated wake

The models presented in section 2.1 are applied to the flight of the Northern Bald Ibis. In scenario 1, the bird is at equilibrium and flies using periodic kinematics. The evolution of the angle of attack, the lift coefficient and the vertical and horizontal forces are represented in Fig 3. The spanwise position $y$ in Fig 3 corresponds to the position of the points along the wing, projected on the $y$ axis. The angle of attack $\alpha$ is the local and instantaneous angle of attack at each point, accounting for the velocities induced by the wake. It is used to compute the $C_L$ from the polar data (which is also represented). The force coefficients $C_Z$ and $C_X$ correspond to the distribution of aerodynamic force per unit distance in the spanwise direction, projected

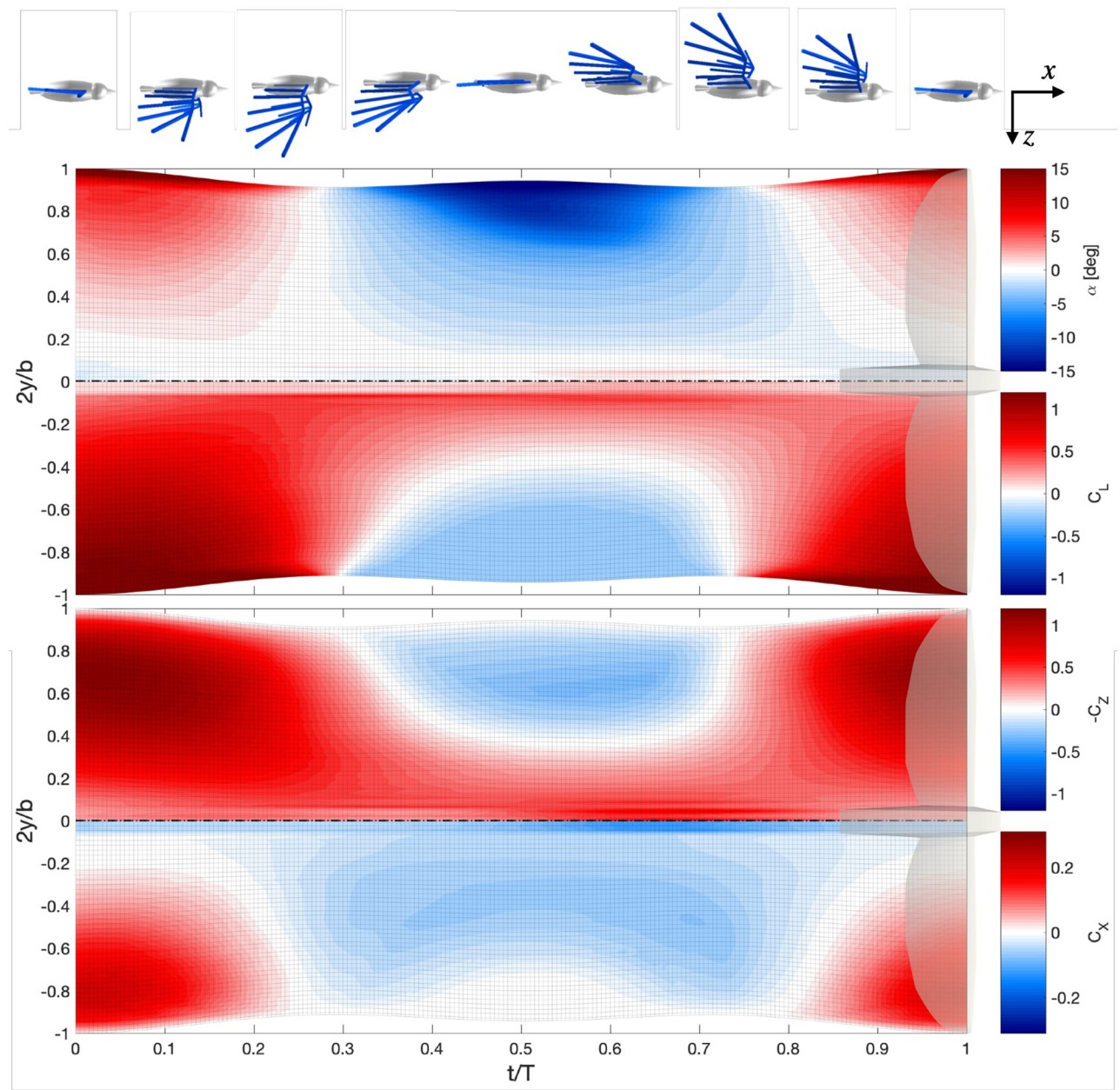

**Fig 3. Time evolution and spanwise distribution of angle of attack $\alpha$, lift coefficient $C_L$ and vertical and horizontal force coefficients ($C_Z$ and $C_X$) through a wingbeat.** Snapshots of the bird during flight are represented above the figure, showing the phase of the flapping motion and the flight direction.

on the $Z$ and $X$ axes, respectively. They are adimensionalized as follows:

$$C_Z(t, y) = \frac{f_Z(t, y)}{\frac{1}{2}\rho U_\infty^2 \bar{c}} \quad,$$

where $f_Z$ is the force per unit spanwise distance. The total force coefficient $C_{Z,tot}$ can be

obtained by integration of this distribution:

$$C_{Z,tot}(t) = \frac{1}{b(t)} \int_{-b(t)/2}^{b(t)/2} C_Z(t,y) \, \mathrm{d}y \ ,$$

where $b(t)$ is the projection of the chord of the bird on the $y$ axis at the time $t$.

In Fig 3, we observe strong variations of the aerodynamic forces over the wingbeat. In downstroke, the angle of attack and the lift coefficient are positive on the whole wing, and increase towards the wingtips. The vertical force has a similar distribution to the one of the lift, but falls at the wingtip as the local chord tends to zero. The horizontal thrust shows that thrust is produced in the downstroke, in the distal parts of the wing. The upstroke has mostly negative angle of attack because of the flapping motion. As the polar has a positive lift coefficient at zero angle of attack, the lift and vertical force coefficients are positive near the body, and negative near the tip. Most of the wing produces drag in the upstroke, but the negative lift combined with the upwards motion of the wing produces slightly positive thrust in the wingtip region.

The fluid solver enables the computation of the wake behind a flying bird. This allows evaluating how much the wake deforms as it is advected downstream. Fig 4 compares the

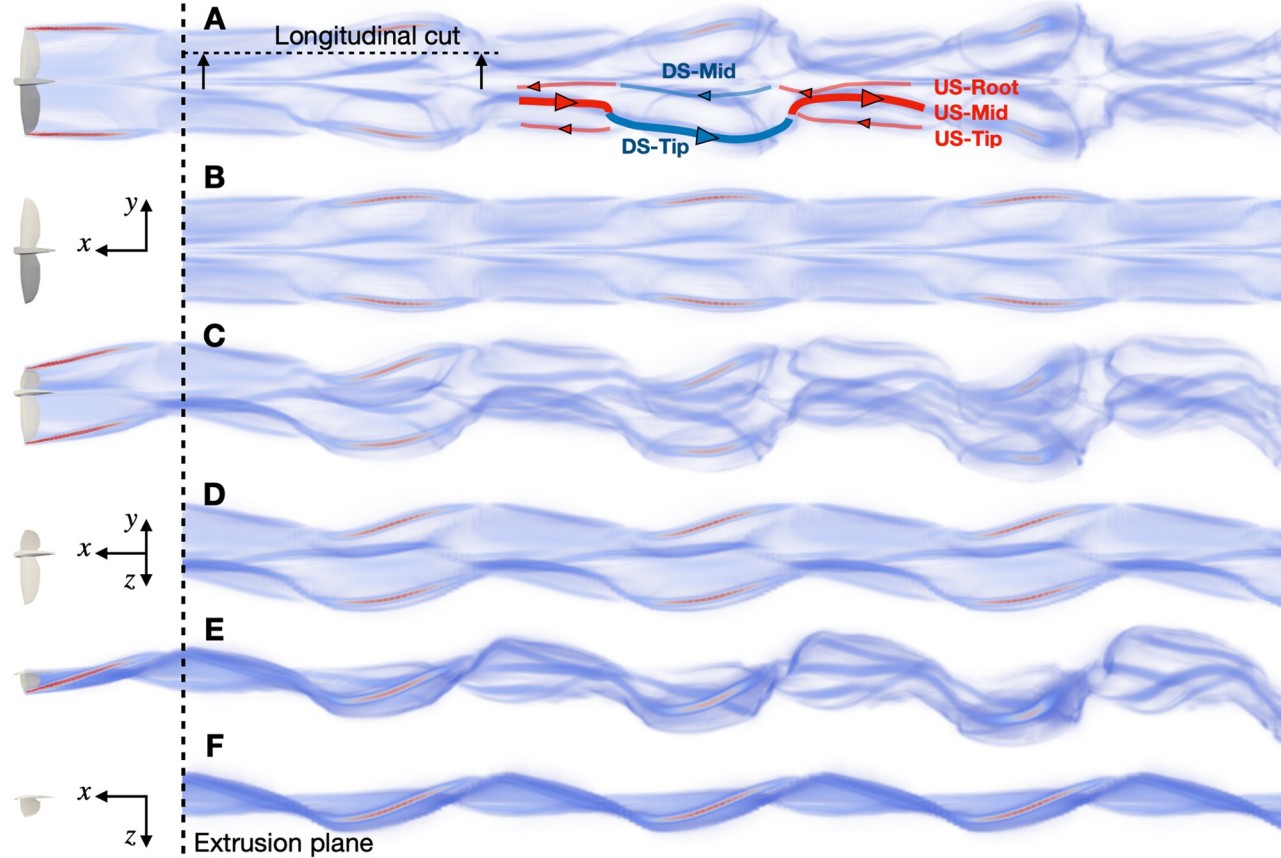

**Fig 4. Scenario 1: Volume rendering of the norm of the vorticity field in the wake of a bird (flying from right to left).** Panels A and B show a top view, C and D an oblique view and E and F a side view. Panels A, C, E represent the instantaneous field at a certain point in the flapping cycle. Panels B, D, F illustrate what can be measured by a single outflow plane by 'extruding' the 2D. In panel A, we added a sketch of the main vortical structures of the wake. The red lines of the sketch represent structures shed during the upstroke, and the blue lines during the downstroke.

instantaneous 3D vorticity field obtained in scenario 1 with an *extruded* wake. The latter is obtained by measuring the field in a 2D surface called the *extrusion plane* positioned at a distance of 10 chords ($10 \cdot \bar{c} = 1.8$ m) behind the bird, and building a 3D image of the wake by substituting the streamwise position with the product of time and free-stream velocity. This wake corresponds to what the frozen-wake assumption would predict.

The structure of the wake observed in Fig 4 corresponds to a strong continuous vortex and several weaker ones. During downstroke, the wing has a high lift, and displays a spanwise gradient of circulation due to the flapping motion. This results in a strong tip vortex (DS-Tip in Fig 4), due to the drop of circulation at the wingtip, and a weaker mid-wing vortex (DS-mid) coming from the vortex sheet resulting from the circulation gradient. The fact that the mid-wing vortex is weaker than the tip vortex indicates that the body has a non-zero circulation, and produces lift. The tip vortex of the downstroke connects to the mid-wing vortex of the upstroke (US-mid). The latter results from the circulation gradient, which changes sign during the upstroke. The tip vortex of the upstroke (US-tip) indicates that the tip of the wing produces negative lift mid-upstroke—as observed in Fig 3—and thus creates a reversed vortex loop with the mid-wing vortex.

The wake deforms as it is advected downstream. The vortex sheet produced by the bird first rolls up into distinct vortices, then these deform under the influence of the velocities they induce on each other. The comparison with the extruded wake shows that, while the main structures of the wake remain identifiable, these deformations are important and lead to approximations if relying on a 2D analysis of the wake.

More details can be seen through 2D visualizations of parts of the wake. Fig 5 shows the spanwise vorticity ($\omega_y$) on longitudinal cuts at various spanwise positions, 10 chords behind the bird wing, as illustrated in panel A of Fig 4. We add a representation of opposite of the time derivative of the circulation in Fig 5. As this quantity is related to the shedding of spanwise vortices to compensate the variations of bound vorticity, we found relevant to represent the opposite of the derivative, since it will thus have the same sign as the spanwise component of the vorticity field.

On the left of Fig 5, we observe that the wing circulation increases at the start of the downstroke on the whole wing, and that it decreases at the transition between downstroke and upstroke, indicating that the wing produces the most lift during downstroke. The body circulation does not follow this trend, and its variations have a lower amplitude. Comparing both sides of Fig 5, we observe that over the distance crossed by the vortical structures, they have deformed and have different shapes then the motion of the wing shedding them. The deformations also lead to the apparitions of additional vortical structures (visible on the second and third panels mainly).

Fig 6 shows the streamwise component of the vorticity field $\omega_x$, with vectors representing the velocity in the plane. The velocity and vorticity fields are measured 10 chords ($10 \cdot \bar{c} = 1.8$ m) behind the bird during scenario 1 (periodic and steady flight). We notice that the streamwise vorticity is lower at the beginning of the downstroke than in the other phases. This implies that the circulation of the wing reaches a minimum around that time. This confirms the results observed in Fig 5. During the downstroke, the strong tip and mid-wing vortices are easily identified. In the upstroke, the tip vortex changes sign, and a third vortex appears in the center of the wing due to the gradient of circulation.

### 3.2 Application of lift estimation models

Wake data obtained through our simulations for scenarios 1 and 2 (and presented in section 3.1 for scenario 1) can be used to evaluate the validity of the expression found in section 3.2,

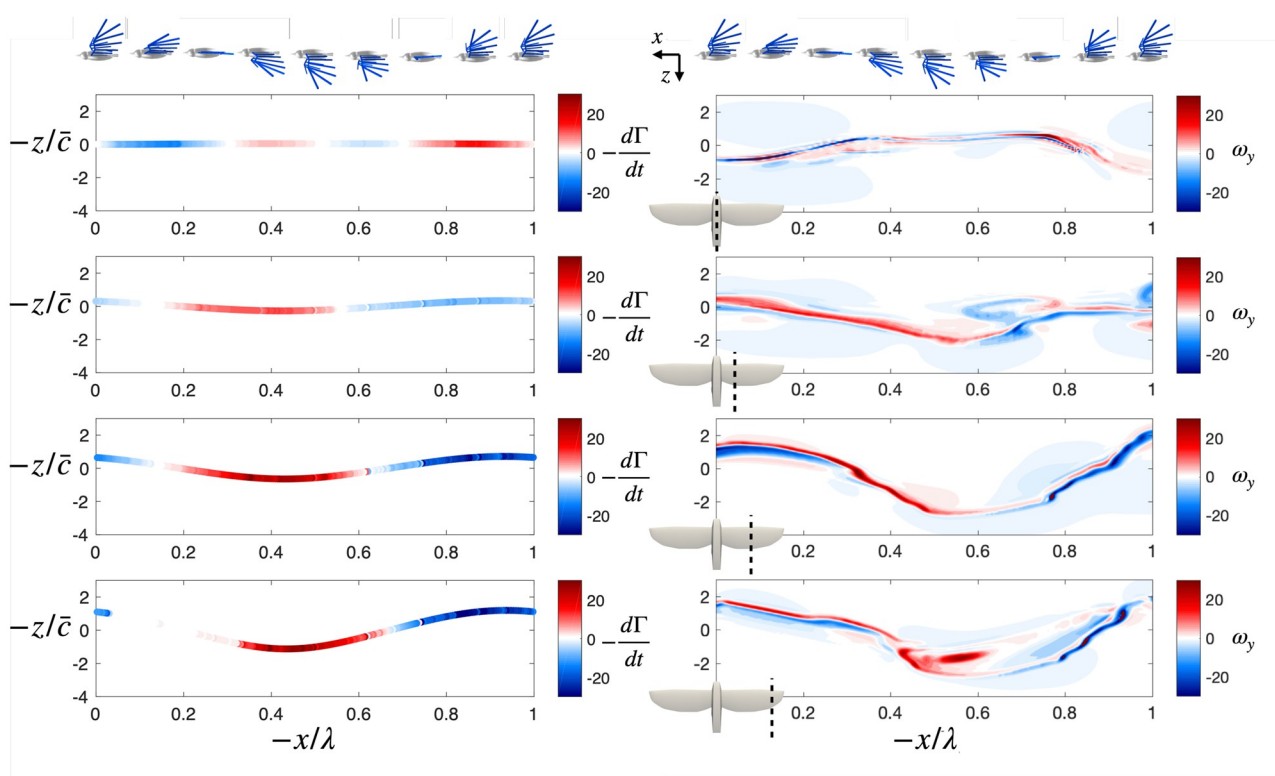

**Fig 5. Scenario 1: Spanwise component of the vorticity field at various positions behind the bird (on the right) and time derivative of the wing circulation at that spanwise position (on the left), represented by color points at the successive locations of the wing.** In both cases, the bird is flying from right to left, and on the right, $x = 0$ corresponds to a distance of 10 chords behind the bird.

and compare them with reference values obtained through volume control analysis. The total vertical aerodynamic force acting on the bird is shown in Fig 7a. The start of the represented flapping period corresponds to the middle of the downstroke. The exact lift based on the aerodynamic performances of the ILDL is compared with the evaluation of Eq (4), showing good agreement. Although the added-mass effect is not accounted for in the simulation, an *a posteriori* estimation of the related contribution, using Eq (1), is also represented in Fig 7a.

We observe from Fig 7b that the simple evaluation of the momentum flux on a closed surface around the bird fails to recover the correct lift force—even with a delay. The term corresponding to the variations of momentum in the volume averages to zero in the case of periodic flight, but the one corresponding to the external forces (e.g., the variations of pressure on the boundaries) does not and contributes to the average lift.

Fig 8 shows the time evolution of the terms in Eq (5) over a time corresponding to one flapping period and in four different outflow planes situated at increasing downstream distances. The terms of the equation are evaluated with the unfiltered velocity and vorticity fields obtained in scenario 1.

Although the terms of Eq (5), which are the ones used in Fig 8, derive from the momentum flux on the outflow plane, we observe in Fig 8 that the estimation provided by Eq (5) has similar values for the extrema and globally the same shape as the actual lift, which was not the case for the results pictured in Fig 7. The quality of the estimation also degrades when the outflow plane moves further downstream from the bird. The phase information is not recovered when using Eq (5), and the phase difference between the estimation and the actual lift varies with the distance.

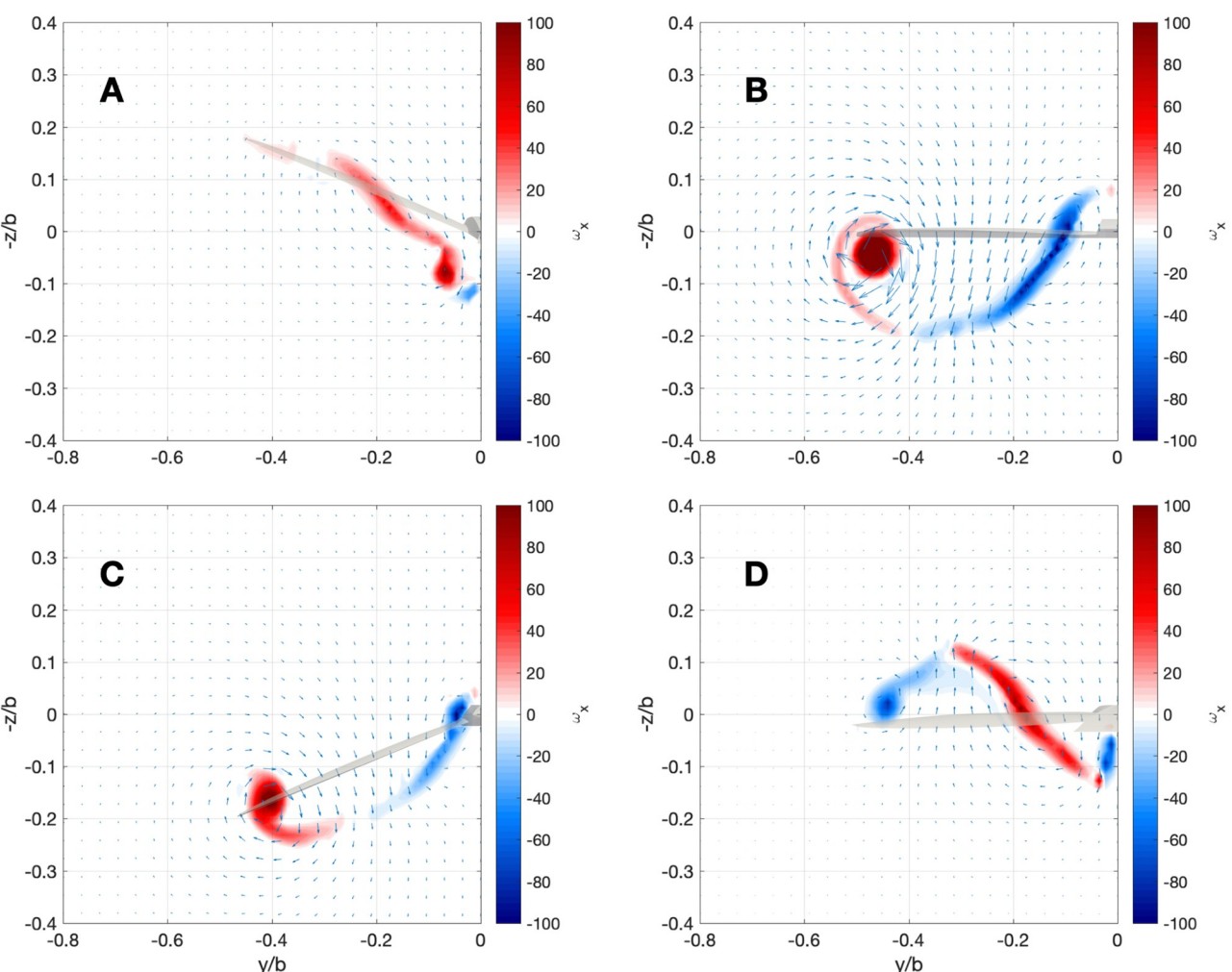

**Fig 6. Scenario 1: Four snapshots of the streamwise component of the vorticity field in a cross flow plane 10 chords behind the bird.** The observed field corresponds to the wake produced when the wing is in the topmost position (A), mid-downstroke (B), at its downmost position (C), and mid-upstroke (D). The arrows represent the velocity in the plane.

We compute the delay as a function of the distance between the bird and the outflow plane, as explained in 1.3. Performing a linear regression on the values of the delay for several distances, we obtain a slope that is almost exactly equal to the free stream velocity, with a relative difference of $8.5710^{-5}$. Fig 9 shows the result of the time-shifting of the $KJ$ estimations of Fig 8.

The evolution of the average lift with the distance is also evaluated. As discussed in section 2.2, the time derivative term then drops. Fig 10 thus shows how the $KJ$ estimation of the period-averaged lift varies with the position of the outflow plane and how the $VD$ term affects it.

From the results shown in Fig 10, we see that estimating the average lift of a flapping flier using only the $KJ$ term provides a good approximation, with an error of less than 3% up to ten spans downstream. The $VD$ term does not contribute much in the estimation and slightly increases the error of the estimation. This is, however, performed in a simulation without noise, so we expect the error to be increased in actual experimental setups.

All the previous results are obtained in the setup provided by scenario 1, that is, for a periodic steady flight. Fig 11 shows an estimation of the lift in scenario 2 (see section 3.3) with non

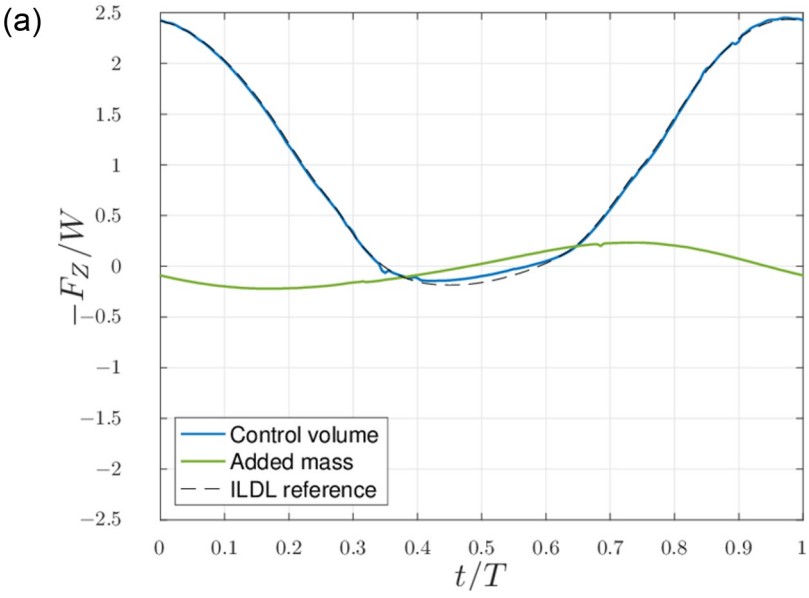

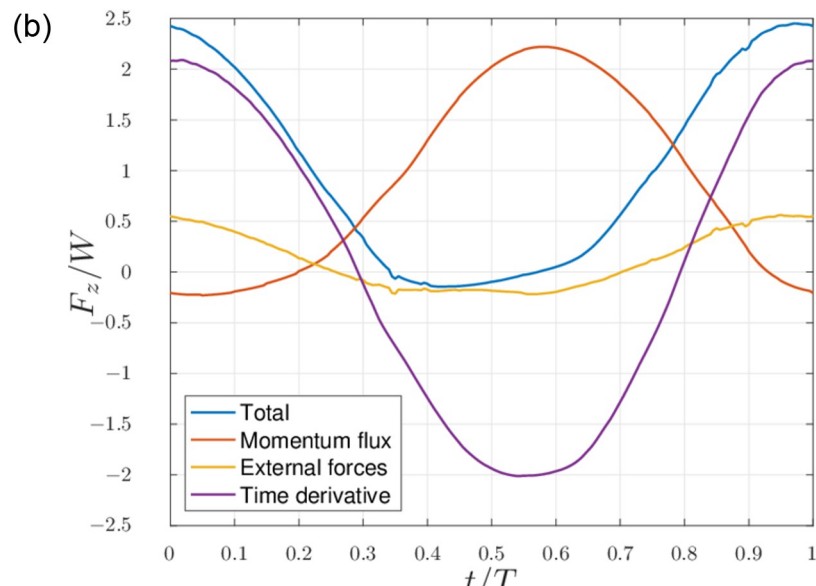

**Fig 7.** Scenario 1: Subfigure (a): total vortex lift (adimensionalized with the weight *W* of the bird) obtained through the ILDL (dashed black) and using Eq (4) (blue), compared to an estimation of the added mass force (green). Subfigure (b): decomposition of the terms of Eq (4) into the corresponding terms of Eq (3).

periodic flapping cycles. The estimation obtained with the *KJ* term only is shifted through time with a delay $\Delta t = x_{out}/U_\infty$. Both the unsteady and period-averaged lift are estimated in three different outflow planes situated at $x_{out}/b = 1, 4$ and $7$, and compared with the lift obtained from the ILDL.

Fig 11 shows that the error in the estimation of the lift increases with the distance between the bird and the outflow plane. The estimation of the unsteady lift reaches higher maximum values for increasing $x_{out}$, and the average of the estimation is also higher than the reference provided by the ILDL-based aerodynamic performances by approximately 5%. However,

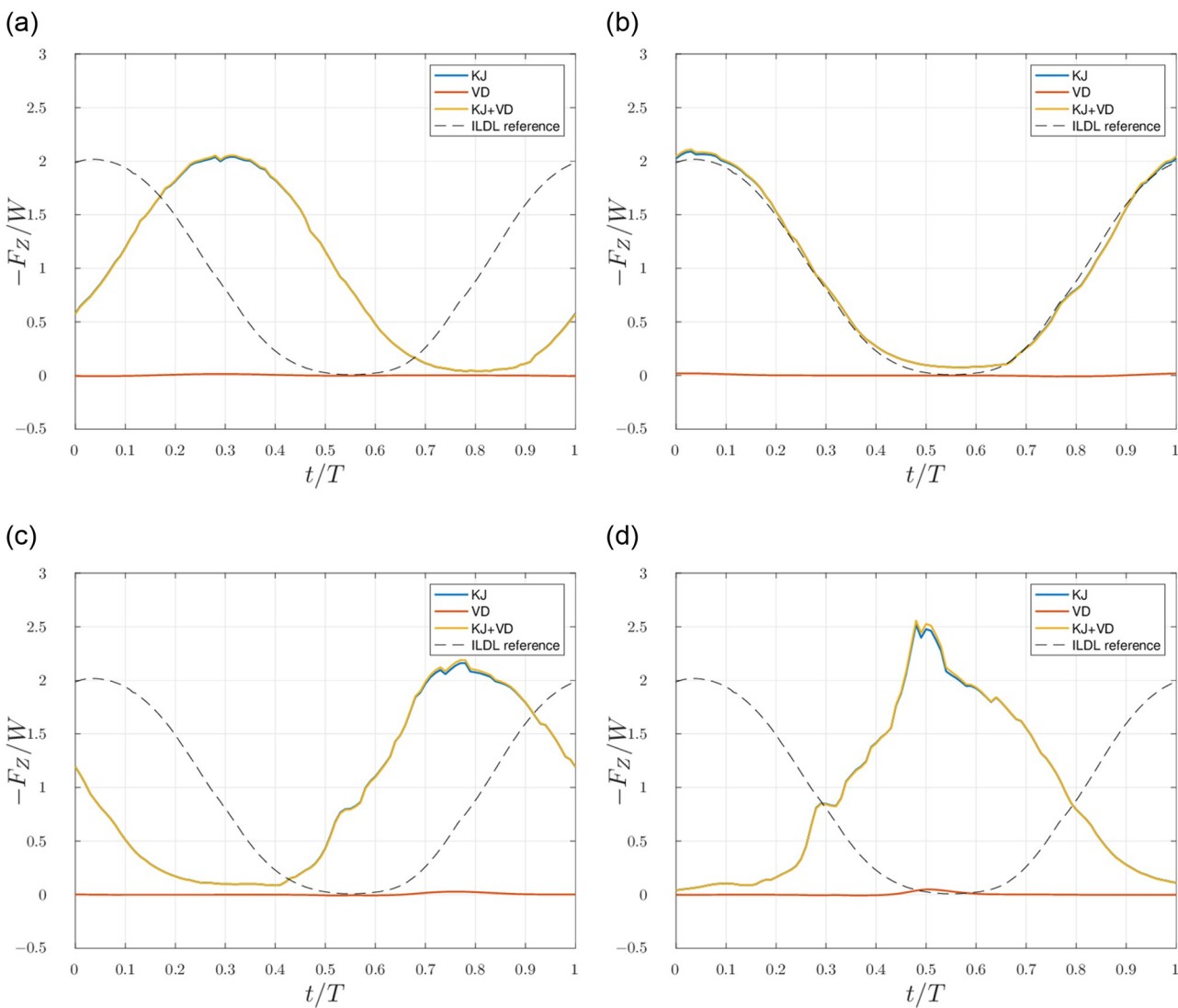

**Fig 8. Scenario 1: Evaluation of both terms of the lift estimation over time for several downstream positions $x_{out}$ of the outflow plane. (a)** $x_{out}/\lambda = 0.25$. **(b)** $x_{out}/\lambda = 1$. **(c)** $x_{out}/\lambda = 1.75$. **(d)** $x_{out}/\lambda = 2.5$.

despite the fact that the lift is not periodic, the phase information is correctly recovered with a time shift of $\Delta t = x_{out}/U_\infty$.

# 4 Discussion

## 4.1 Simulated bird wake

The simulation framework allows reproducing the wake of a flying and flapping bird as if it was observed in a wind tunnel. The simulated wake shows similar structures to that of an actual bird. The most notable features are strong tip vortices, an expected feature in flapping or finite wing flight wakes [8, 22, 30]. The wake also presents root vortices that have been observed in [8, 9, 22] and spanwise vortices due to time variations of the wing circulation, as in [4, 11]. During the upstroke, the tip vortex changes sign, and a third vortex appears at a mid-span location along the wing. While this last phenomenon is not usually observed in bird

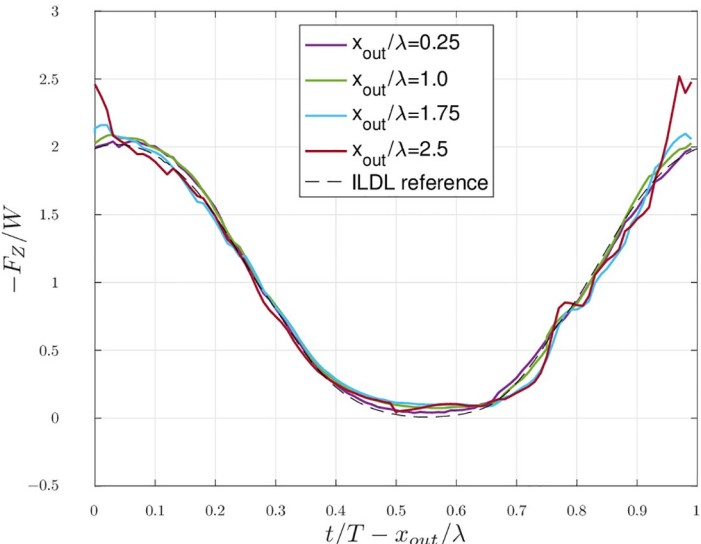

**Fig 9. Scenario 1: *KJ* estimations of the instantaneous lift for several positions of the outflow plane $x_{out}$, shifted in time with $\Delta t = x_{out}/U_\infty$.**

flight, it has been in bat flight [9]. The important differences between the wake extruded from a single measurement plane and the actual 3D wake (see Fig 4) indicate that the deformations of the wake are not negligible and that the frozen-wake assumption only holds for small streamwise distances.

The spanwise component of the vorticity field presented in Fig 5 has two origins. First, it is the wake signature of the profile drag. The ILDL method creates dipoles of particles that represent the velocity deficit created by this force. This results in the two layers of opposite sign vorticity. The other origin of spanwise vorticity is the time variations of the circulation illustrated

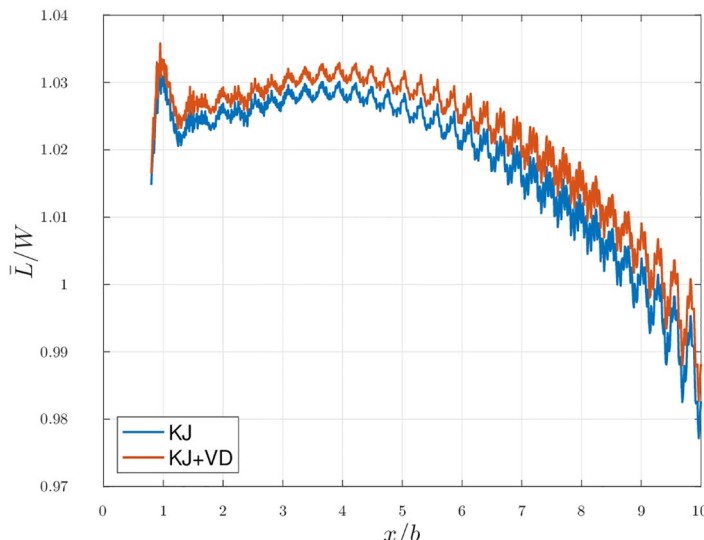

**Fig 10. Scenario 1: Average the lift estimation as a function of the downstream distance.**

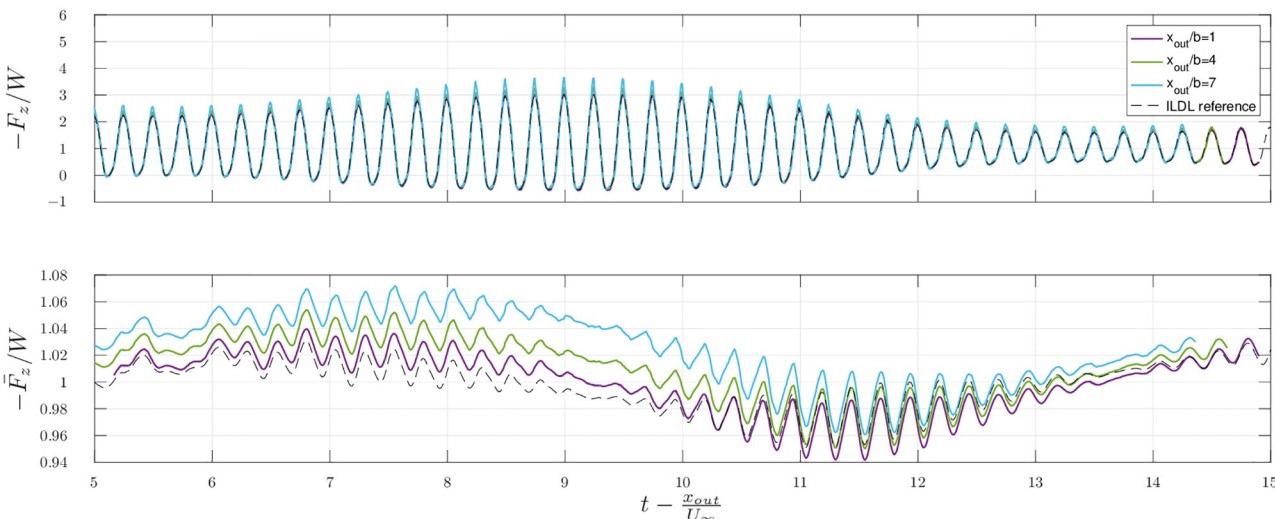

**Fig 11. Scenario 2: Estimation of the lift for non-periodic flapping cycles.** The upper figure shows the estimations of the instantaneous lift and the lower one of the lift averaged over the previous flapping period ($\bar{F}_z(t) = \frac{1}{T}\int_{t-T}^{t} F_z \, dt'$), for several distances $x_{out}$ of the outflow plane.

in the same figure. Since the drag is represented as dipoles of particles of opposite intensity, the average circulation of the sheet at a given position is determined by the time variations of the circulation.

The spanwise vorticity field obtained in the simulations can be compared to experimental results where this field has been measured. In particular, we compare our results to the ones obtained for a swift in [5] (which also has a high aspect ratio wing). We refer the reader to Fig 4 of this paper, which is similar to how we presented Fig 5. Comparing both figures, we see that the wake produced by our simulations is very similar to the one of an actual bird. In [5], we can also observe the two layers of vorticity corresponding to the drag, and the variations of intensity of the vortex sheet corresponds to our results, as the circulation of the wing increases at the start of the downstroke, then decreases at the start of the upstroke. The wake of the body obtained in simulations, however, shows significant differences with the experimental data.

Measurements of spanwise vorticity have also been reported in [7, 31, 32]. In both [7, 31], the signature of drag is well visible at the considered spanwise position. In [32], more complex structures are observed. It is a possibility that the multiple layers of vorticity observed in Fig 5 may correspond to the double branch features reported and discussed in [32], as they also occur in regions of transition between strokes, where the variation of circulation is important. However, our model does not include multiple shedding points from the wing (aside from the shedding of drag-related structures), as would be the case in highly unsteady cases. Thus, the origin of these structures, in our simulations, is in the deformations occurring between the bird and the measurement position.

The streamwise component of the vorticity field represented in Fig 6 can also be compared to experimental data. In [8], similar results have been reported for a swift. Comparing our results to the ones found in Fig 3 of [8], we observe similar vortices, with the least intensity at the start of the downstroke, a strong tip vortex and a spread root/mid-wing vortex during the downstroke, and even what could correspond to the opposite sign tip vortex produced during the upstroke in our simulations (in panel (d) of Fig 3 of [8]).

Although the wake studied in our simulations results from a model with multiple assumptions, it is similar to bird wakes observed experimentally. We can thus consider that the methods used to estimate the lift of the bird from measurements of our simulated wake can also apply in experimental studies about birds with high aspect ratio.

## 4.2 Lift estimation

In Fig 7a, we see that the evaluation of the full Eq (4) agrees very well with the force from the lifting line. The added mass, also represented in Fig 7a, is of much lower amplitude than the total lift but is not necessarily negligible. When averaged, though, this contribution becomes negligible compared to the average lift.

As stated before, the momentum flux alone does not provide a good estimation for the total lift. Even when computed on a closed surface around the bird, it results in a force profile that is delayed and lower on average than the actual lift. In the results shown in Fig 7, the control volume has a width equal to 6 times the span of the bird, and the term related to the influence of the pressure is still significant.

Fig 8 shows that the *KJ* term gives an acceptable estimation of the lift—with a delay, as discussed in section 2.2. Although this term is derived from the flux of momentum through the outflow plane, the good agreement between the total force and this estimation implies that it is more accurate to use it than to directly integrate the momentum flux. This is due to the use of vorticity in Eq (5)—which is a compact field—instead of the velocity (like in Eq (4)) in the integrals giving an estimation of the lift. Thus, a finite outflow plane can provide an exact evaluation of the integrals as long as it encompasses the region of non-zero vorticity.

The contribution of the *velocity departure* term to the total lift estimation appears to be negligible. This means that access to the local streamwise velocity in the outflow plane is not necessary to obtain a good estimation of the lift as long as the free stream velocity is known. The fact that the plane of integration does not extend to infinity implies that the evaluation of the term is incomplete (because the velocity does not form a compact field). However, since the contribution of the *VD* term is negligible, this is not an issue.

Wang and his co-authors used the *KJ* term to evaluate the lift of a flapping plate in [14]. Like in the present work, they obtained good results when computing the average lift, in which case the added mass contribution is deemed negligible. This is backed by the similarity of the results for the average lift in this work and in [14]. The use of the vorticity in the outflow plane to estimate the instantaneous lift gave poor results in [14], even when correcting for the delay. These deviations with respect to the actual lift are possibly due to the contribution of added mass or viscosity effects since the Reynolds number they used was relatively low (*Re* = 300). Concerning the former, it can be seen in Fig 12 of [14] that the amplitude of the added mass lift is about twice that of the vortex-related force. The magnitude of the added mass effect relative to the vortical lift is directly related to the reduced frequency *k* (see S3 Appendix). In their case, this parameter was equal to 1.89, while in our simulations, the reduced frequency based on the mean aerodynamic chord of the wing is only 0.15. Therefore, it is expected that the added mass effects would be tenfold more influential in their setting than in ours. Concerning the effect of viscosity, its importance is inversely proportional to the Reynolds number, and it increases the streamwise diffusion of vertical momentum. The time-averaged momentum flux through an outflow plane is conserved, but its diffusion in the streamwise direction dampens the oscillations in the estimations of the lift, as clearly revealed by Figs 2, 9 and 16 of [14]. Since our simulations are performed at a higher Reynolds number ($Re \simeq 10^5$), this streamwise diffusion occurs more slowly and is actually overwhelmed by the self-induced distortions of the wake. Therefore, the unsteady lift can be recovered for further positions of the outflow

plane before the distortions affect the instantaneous estimation. The framework of the present work is thus more adapted to the study of flight of large birds, which is characterized by lower values of the reduced frequency and higher Reynolds numbers.

In our simulations, we compute the delay between the instantaneous lift and the estimation obtained with the *KJ* term. This delay is related to the time necessary for the vorticity shed by the bird to reach the outflow plane. While this delay is likely directly related to the forward flight velocity of the bird $V$ and evolving as $\Delta t = x_{out}/V$, it may be affected by the streamwise velocities induced by the bird and its wake. The fact that this delay evolves almost exactly as $\Delta t = x_{out}/U_\infty$ is linked to the horizontal equilibrium of the bird. Even though profile drag induces a velocity deficit in the wake, the wings produce an equal thrust force, accelerating the fluid. However, if the bird has a non-zero horizontal acceleration, its exchanges of momentum with the fluid may lead to the vorticity in the wake traveling faster or slower, and thus to an effect on the delay of the estimation.

In the second considered scenario, the bird generates non periodic wing kinematics, leading successively to an increase and a decrease of the lift before settling to the initial value. We saw in this case that the variations of lift were smaller than expected. The flight dynamics indeed lead to vertical velocities of the bird that tend to reduce the variations of lift by affecting the angle of attack. In this case, Fig 11 shows that the time shift corresponds to the time needed for the wake to travel to the measurement plane. Any changes in the average velocity at which the wake travels downstream are negligible here because the horizontal accelerations of the bird are very small.

Although the quality of the estimation remains acceptable at a distance of seven spans, we see that it degrades as the integration plane moves further downstream from the bird. This is due to growing distortion in the wake, leading to different travel times for the vortical structures. The average of the estimation shows a larger departure from the ground truth in this scenario than in the stationary case. The error increases as the distance between the bird and the outflow plane increases. To avoid imprecisions in the estimation of both the instantaneous and averaged lift, as expected, the estimations are best performed at the closest possible position downstream from the bird.

## 5 Conclusion

In this paper, lift estimation techniques are applied to the simulation of a large bird in flapping flight. We use a numerical framework simulating the flight of a bird to model the wake behind it. The wake observed during the numerical experiments presents the same main features as what can be measured experimentally [4, 5, 8, 11, 32]. In particular, we compared the structure and evolution of the wake to the one of a swift, another high-aspect ratio bird, presented in [5, 8], showing that the vorticity field observed in the simulation was very similar to the one behind a real bird.

The advantage of the numerical approach is that it provides access to the velocity and vorticity fields everywhere around the bird. It notably allows the evaluation of exact expressions of its lift based on velocity measurements, such as in Eq (4). The result of the lift computed through this equation shows that we can indeed compute the instantaneous aerodynamic force provided that the velocity is accessible all around a bird. However, as discussed in the introduction, experimental works are more frequently based on measurements in a single plane in the wake of the bird. We have shown that it is possible to estimate the instantaneous lift of a flapping bird using vorticity measurements in a cross-flow plane behind it.

We also identify the underlying assumptions of the estimation and quantify them. Added mass is ignored in the estimation, and cannot be retrieved solely from measurements in the

bird wake. The only way to properly evaluate its effect is to measure the velocity all around the wings, but we find that its contribution is small in comparison to the circulatory lift. The ratio between the added mass force and the circulatory lift is approximately a quarter of the reduced frequency $k$, which is small for the studied case of large birds. In order to accurately recover the lift variations, viscous effects and the ensuing diffusion of momentum must be negligible. We find it is the case for a Reynolds number of $10^5$ corresponding to the flight of an ibis. Finally, for the estimation to be accurate, the bird needs to fly at a constant velocity without accelerating.

When these conditions are met, we find that the average of the estimation exhibits remarkable consistency with increasing downstream distances. The proposed formula can also be used to recover the circulation-related contribution of the instantaneous lift—provided that a time delay is accounted for. The estimated time-dependent lift is found to closely match the reference values at close distances from the bird. While the quality of the estimation degrades, it remains acceptable for streamwise distance of up to seven spans.

## Supporting information

**S1 Appendix. Aerodynamic polar used in the simulations.**
(PDF)

**S2 Appendix. Derivation of the lift estimation formula from the momentum flux.**
(PDF)

**S3 Appendix. Ratio of the circulatory to added mass lift forces for a 2D wing in plunging motion.**
(PDF)

## Author Contributions

**Conceptualization:** Victor Colognesi, Renaud Ronsse, Philippe Chatelain.

**Data curation:** Victor Colognesi.

**Formal analysis:** Victor Colognesi, Renaud Ronsse, Philippe Chatelain.

**Investigation:** Victor Colognesi, Renaud Ronsse, Philippe Chatelain.

**Methodology:** Victor Colognesi, Renaud Ronsse, Philippe Chatelain.

**Project administration:** Renaud Ronsse, Philippe Chatelain.

**Resources:** Renaud Ronsse, Philippe Chatelain.

**Software:** Victor Colognesi, Philippe Chatelain.

**Supervision:** Renaud Ronsse, Philippe Chatelain.

**Visualization:** Victor Colognesi.

**Writing – original draft:** Victor Colognesi.

**Writing – review & editing:** Victor Colognesi, Renaud Ronsse, Philippe Chatelain.

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
