## [Decision Letter · Decision Letter 0]

4 Dec 2022

PONE-D-22-25640Numerical assessment of wake-based estimation of instantaneous lift in flapping flightPLOS ONE

Dear Dr. Colognesi,

Thank you for submitting your manuscript to PLOS ONE. After careful consideration, we feel that it has merit but does not fully meet PLOS ONE’s publication criteria as it currently stands. Therefore, we invite you to submit a revised version of the manuscript that addresses the points raised during the review process.

We look forward to receiving your revised manuscript.

Kind regards,

Roi Gurka

Academic Editor

PLOS ONE

Journal Requirements:

“Victor Colognesi is supported by a FRIA grant from the Fonds de la Recherche Scientifique de Belgique (F.R.S.- 404 FNRS). 405 This work is part of a project supported by the ARC program of the Federation Wallonie-Bruxelles (grant 406 number 17/22-080, RevealFlight). 14 407 Computational resources have been provided by the Consortium des Equipements de Calcul Intensif (CECI), 408 funded by the Fonds de la Recherche Scientifique de Belgique (F.R.S.-FNRS) under Grant No. 2.5020.11 and by 409 the Walloon Region. 410 The present research also benefited from computational resources made available on the Tier-1 supercomputer 411 of the Federation Wallonie-Bruxelles, infrastructure funded by the Walloon Region under the grant agreement 412 1117545.”

“Victor Colognesi is supported by a FRIA grant (Grant number FC 21291) from the Fonds de la Recherche Scientifique de Belgique (F.R.S.- FNRS, https://www.frs-fnrs.be/en/).

This work is part of a project supported by the ARC program (grant number 17/22-080, RevealFlight) of the Federation Wallonie-Bruxelles (https://www.federation-wallonie-bruxelles.be).

Computational resources have been provided by the Consortium des Equipements de Calcul Intensif (CECI), funded by the Fonds de la Recherche Scientifique de Belgique (F.R.S.-FNRS) under Grant No. 2.5020.11 and by the Walloon Region.

The present research also benefited from computational resources made available on the Tier-1 supercomputer of the Federation Wallonie-Bruxelles, infrastructure funded by the Walloon Region under the grant agreement 1117545.

Reviewers' comments:

Reviewer's Responses to Questions

**Comments to the Author**

1. Is the manuscript technically sound, and do the data support the conclusions?

Reviewer #1: Yes

Reviewer #2: Partly

2. Has the statistical analysis been performed appropriately and rigorously? 

Reviewer #1: Yes

Reviewer #2: Yes

3. Have the authors made all data underlying the findings in their manuscript fully available?

Reviewer #1: Yes

Reviewer #2: Yes

4. Is the manuscript presented in an intelligible fashion and written in standard English?

Reviewer #1: Yes

Reviewer #2: Yes

5. Review Comments to the Author

Reviewer #1: This work introduces a lift estimation technique for large Reynolds number flapping flight. The paper is well written, and the arguments are convincing. However, there are some major revisions needs to be done to improve the quality of this work:

• This study derives and then validates an instantaneous lift equation using numerical simulation data of a flapping flight model. However, no validation data for the numerical solver had been provided. At the least, velocity profiles and vorticity profiles at the Trefftz planes (using the numerical solver) need to be compared with the experimental data. Although the authors provided a reference for the solver details, this comparison needs to be shown in the main manuscript. Without these comparisons with the experimental data, the validity of the solver remains questionable.

• The authors mentioned that AS6092 (bird-like) airfoil was used for the simulations. However, the simulated bird was Geronticus Eremita. This fact largely undermines the validity of this study as a slight difference in camber or curvature (with the actual bird airfoil) would produce significant difference in numerical simulation compared to the experimental data.

• Again, birds’ airfoil can vary significantly along the span in terms of camber and chord length. The authors didn’t mention anything about the differences between the model and the actual bird (Geronticus Eremita). A figure showing the wing model as well as the actual bird wing is required to be added in the manuscript.

• Also, this study lacks content. I would expect the authors to include some contents that show variations of lift as a function of the AOA. Since this is a numerical experiment, this can be done after validating the solver data with the level flight of the bird (experimental data).

• It is not clear to me how mathematical operations for the surface integral of the velocity were done to turn it into a surface integral of vorticity*distance (equation 12 in appendix). Can you please show the step by step processes?

• The figures (4a-5d, 7) need line legend.

• I would recommend authors to add more contents/pictures in Figure 3 (evolution of wake vortices). As this is a numerical simulation, the authors should be able to provide a much more detailed picture of evolution of vorticity at the wake.

• To use the lift equation (equation 5), is there any requirement to use threshold on the vorticity maps? Did the authors use any threshold on Figure 3?

Reviewer #2: Dear authors,

I have attached my comments on your manuscript (PONE-D-22-25640) in two .pdf files.

The first .pdf file contain specific comments, and the second .pdf file consists of some minor grammar mistakes and typos to address.

6. PLOS authors have the option to publish the peer review history of their article (what does this mean?). If published, this will include your full peer review and any attached files.

Reviewer #1: **Yes: **Asif Shahriar Nafi

Reviewer #2: **Yes: **Hadar Ben-Gida

---

## [Author Response · Author response to Decision Letter 0]

1 Feb 2023

We thank the reviewers for all their comments and careful revision of our work. We addressed all the points raised by both reviewers. We uploaded the revised manuscript in two versions: a clean one and a marked-up one. All the comments are answered individually in the response_to_reviewers file that we uploaded.

---

## [Decision Letter · Decision Letter 1]

14 Mar 2023

PONE-D-22-25640R1Numerical assessment of wake-based estimation of instantaneous lift in flapping flight of large birdsPLOS ONE

Dear Dr. Colognesi,

Thank you for submitting your manuscript to PLOS ONE. After careful consideration, we feel that it has merit but does not fully meet PLOS ONE’s publication criteria as it currently stands. Therefore, we invite you to submit a revised version of the manuscript that addresses the points raised during the review process.

We look forward to receiving your revised manuscript.

Kind regards,

Roi Gurka

Academic Editor

PLOS ONE

Journal Requirements:

Reviewers' comments:

Reviewer's Responses to Questions

**Comments to the Author**

1. If the authors have adequately addressed your comments raised in a previous round of review and you feel that this manuscript is now acceptable for publication, you may indicate that here to bypass the “Comments to the Author” section, enter your conflict of interest statement in the “Confidential to Editor” section, and submit your "Accept" recommendation.

Reviewer #1: All comments have been addressed

Reviewer #2: All comments have been addressed

2. Is the manuscript technically sound, and do the data support the conclusions?

Reviewer #1: Yes

Reviewer #2: Yes

3. Has the statistical analysis been performed appropriately and rigorously? 

Reviewer #1: N/A

Reviewer #2: Yes

4. Have the authors made all data underlying the findings in their manuscript fully available?

Reviewer #1: Yes

Reviewer #2: Yes

5. Is the manuscript presented in an intelligible fashion and written in standard English?

Reviewer #1: Yes

Reviewer #2: Yes

6. Review Comments to the Author

Reviewer #1: (No Response)

Reviewer #2: (No Response)

7. PLOS authors have the option to publish the peer review history of their article (what does this mean?). If published, this will include your full peer review and any attached files.

Reviewer #1: No

Reviewer #2: **Yes: **Hadar Ben-Gida

---

## [Editor Report · Decision Letter 2]

6 Apr 2023

Numerical assessment of wake-based estimation of instantaneous lift in flapping flight of large birds

PONE-D-22-25640R2

Dear Dr. Colognesi,

We’re pleased to inform you that your manuscript has been judged scientifically suitable for publication and will be formally accepted for publication once it meets all outstanding technical requirements.

Kind regards,

Roi Gurka

Academic Editor

PLOS ONE
---

## [Editor Report · Acceptance letter]

25 Apr 2023

PONE-D-22-25640R2 

Numerical assessment of wake-based estimation of instantaneous lift in flapping flight of large birds  

Dear Dr. Colognesi:

I'm pleased to inform you that your manuscript has been deemed suitable for publication in PLOS ONE. Congratulations! Your manuscript is now with our production department. 

Kind regards, 

on behalf of

Dr. Roi Gurka 

Academic Editor

PLOS ONE